# Srebp-1c/Fgf21/Pgc-1α Axis Regulated by Leptin Signaling in Adipocytes—Possible Mechanism of Caloric Restriction-Associated Metabolic Remodeling of White Adipose Tissue

**DOI:** 10.3390/nu12072054

**Published:** 2020-07-10

**Authors:** Masaki Kobayashi, Seira Uta, Minami Otsubo, Yusuke Deguchi, Ryoma Tagawa, Yuhei Mizunoe, Yoshimi Nakagawa, Hitoshi Shimano, Yoshikazu Higami

**Affiliations:** 1Laboratory of Molecular Pathology and Metabolic Disease, Faculty of Pharmaceutical Sciences, Tokyo University of Science, Chiba 278-8510, Japan; kobayashim@rs.tus.ac.jp (M.K.); 3B18508@ed.tus.ac.jp (S.U.); 3a15020@ed.tus.ac.jp (M.O.); 3B16060@ed.tus.ac.jp (Y.D.); tagawar@rs.tus.ac.jp (R.T).; 2Department of Internal Medicine (Endocrinology and Metabolism), Faculty of Medicine, University of Tsukuba, Ibaraki 305-8575, Japan; ymizunoe@md.tsukuba.ac.jp (Y.M.); hshimano@md.tsukuba.ac.jp (H.S.); 3Division of Complex Biosystem Research, Department of Research and Development, Institute of Natural Medicine, University of Toyama, Toyama 930-0194, Japan; ynaka@inm.u-toyama.ac.jp; 4Life Science Center for Survival Dynamics, Tsukuba Advanced Research Alliance (TARA), University of Tsukuba, Ibaraki 305-8575, Japan; 5AMED-CREST, Japan Agency for Medical Research and Development (AMED), Tokyo 100-1004, Japan; 6Research Institute for Biomedical Sciences, Tokyo University of Science, Chiba 278-8510, Japan

**Keywords:** caloric restriction, fatty acid biosynthesis, mitochondrial biogenesis, adipocyte

## Abstract

Caloric restriction (CR) improves whole body metabolism, suppresses age-related pathophysiology, and extends lifespan in rodents. Metabolic remodeling, including fatty acid (FA) biosynthesis and mitochondrial biogenesis, in white adipose tissue (WAT) plays an important role in the beneficial effects of CR. We have proposed that CR-induced mitochondrial biogenesis in WAT is mediated by peroxisome proliferator-activated receptor γ coactivator-1α (PGC-1α), which is transcriptionally regulated by sterol regulatory element-binding protein 1c (SREBP-1c), a master regulator of FA biosynthesis. We have also proposed that the CR-associated upregulation of SREBP-1 and PGC-1α might result from the attenuation of leptin signaling and the upregulation of fibroblast growth factor 21 (FGF21) in WAT. However, the detailed molecular mechanisms remain unclear. Here, we interrogate the regulatory mechanisms involving leptin signaling, SREBP-1c, FGF21, and PGC-1α using *Srebp-1c* knockout (KO) mice, mouse embryonic fibroblasts, and 3T3-L1 adipocytes, by altering the expression of SREBP-1c or FGF21. We show that a reduction in leptin signaling induces the expression of proteins involved in FA biosynthesis and mitochondrial biogenesis via SREBP-1c in adipocytes. The upregulation of SREBP-1c activates PGC-1α transcription via FGF21, but it is unlikely that the FGF21-associated upregulation of PGC-1α expression is a predominant contributor to mitochondrial biogenesis in adipocytes.

## 1. Introduction

It is well known that white adipose tissue (WAT) is involved in the pathogenesis of age-related diseases including type 2 diabetes, atherosclerosis, and other cardiovascular and cerebrovascular diseases [1]. It has recently been shown that WAT quality, including adipocyte size, mitochondrial biogenesis, and adipokine expression profile, is a key player in lifespan regulation [2,3,4,5,6].

Caloric restriction (CR) is the most robust, reproducible, and simple experimental manipulation that is capable of improving whole body metabolism, delaying the onset of various age-related pathophysiological changes and extending both median and maximum lifespan in a wide range of organisms [7,8]. Dwarf rodents that demonstrate the suppression of growth hormone/insulin-like growth factor 1 (GH/IGF-1) signaling live longer than their wild-type (Wd) littermates [9]. Since CR suppresses GH/IGF-1 signaling, its beneficial effects are considered to be dependent on the suppression of GH/IGF-1 signaling [10]. However, CR further extends the lifespan of long-lived dwarf rodents that have GH/IGF-1 suppression [11,12]. Therefore, the beneficial effects of CR are also likely to be mediated through a GH/IGF-1-independent mechanism.

To identify the GH/IGF-1-independent mechanism involved in the effects of CR, we compared the gene expression profile of the WAT of long-living dwarf rats bearing an antisense GH transgene with that of Wd rats subjected to CR, and we found that CR upregulated the expression of genes involved in fatty acid (FA) biosynthesis in a GH/IGF-1-independent manner [13]. Sterol regulatory element binding protein-1 (SREBP-1), including its two isoforms, SREBP-1a and -1c, is a master transcriptional regulator of FA biosynthesis [14]. In WAT, SREBP-1c is predominantly expressed, rather than SREBP-1a [15]. Therefore, we applied CR to both *Srebp-1c* knockout (KO) and WT mice on a B6; 129S6 background and found that CR extended lifespan in Wd mice but not in KO mice. Moreover, CR upregulated the expression of proteins involved in FA biosynthesis and mitochondrial biogenesis in the WAT of Wd mice but not in KO mice. These findings were observed only in WAT but not in the other tissues, including liver, kidney, quadriceps femoris muscle, and heart [16]. Peroxisome proliferator-activated receptor γ coactivator-1α (PGC-1α) is a master transcriptional cofactor for mitochondrial biogenesis [17] and a key regulator of the CR-induced activation of mitochondrial biogenesis [18]. We also found that CR upregulates *Pgc-1a* mRNA in WAT of Wd mice but not in *Srebp-1c* KO mice. Moreover, a chromatin immunoprecipitation assay showed that SREBP-1 protein binds to the promoter region of the *Pgc-1a* gene, as well as the *Fasn* gene, in mouse embryonic fibroblasts (MEFs) derived from WT mice but not in those from KO mice. Therefore, we suggested that CR upregulates FA biosynthesis and mitochondrial biogenesis via SREBP-1c in WAT [16].

Fibroblast growth factor 21 (FGF21), which was initially identified as a hepatokine, is mostly secreted by the liver [19]. Circulating FGF21 binds to the FGF receptor (FGFR) and β-klotho (KLB) receptor complex in target tissues such as WAT. The binding of FGF21 to its receptors activates downstream signaling, including extracellular signal-regulated kinase (ERK) signaling, which upregulates the expression of genes involved in glucose and lipid metabolism [20,21,22]. FGF21 expression is negatively regulated by SREBP-1c in hepatocytes [23]. In contrast, FGF21 expression is upregulated by SREBP-1c in WAT and 3T3-L1 adipocytes [24]. FGF21 induces PGC-1α expression in the liver as an adaptation to starvation [25]. In WAT, FGF21 positively regulates PGC-1α and PPARγ expression and/or activity via feed-forward autocrine/paracrine loops [26,27]. Moreover, *Fgf21* Tg mice live longer than Wd mice and have a similar metabolic phenotype to CR mice [28]. We have also shown that the CR-associated upregulation of PGC-1α expression is partially mediated through FGF21 in WAT [29]. CR also upregulates PPARγ expression in WAT [29]. In addition, the expression of PGC-1α is increased as a result of rosiglitazone-induced PPARγ activity in WAT [30].

Leptin, which was the first substance to be identified as an adipokine, is mostly secreted by adipocytes [31]. Circulating leptin binds to the leptin receptor, which is predominantly expressed in the arcuate nucleus of the hypothalamus and reduces appetite and increases energy expenditure via the sympathetic nervous system [32]. However, the leptin receptor is also expressed in other cell types, including adipocytes [33]. It has been reported that leptin treatment downregulates the expression of SREBP-1 and its downstream targets in mouse WAT [34]. In addition, CR reduces leptin secretion by adipocytes, thereby reducing the circulating leptin concentration [35]. These findings raised the possibility that CR might suppress leptin signaling via an autocrine/paracrine loop, leading to the SREBP-1-induced upregulation of proteins involved in FA biosynthesis in WAT.

As stated above, the molecular mechanisms of CR-associated metabolic remodeling, including FA biosynthesis and mitochondrial biogenesis, are unclear. In particular, the reciprocal regulatory mechanism that involves SREBP-1, FGF21, and PGC-1α is complex. In the present study, we aimed to clarify this molecular mechanism, focusing on the expression of the master regulators of FA biosynthesis and mitochondrial biogenesis, SREBP-1 and PGC-1α, respectively, in adipocytes. To this end, we analyzed the regulation of leptin signaling, SREBP-1c, FGF21, and PGC-1α in the CR-associated metabolic remodeling of WAT and adipocytes.

## 2. Materials and Methods

### 2.1. Animals and the Collection of Mice Embryonic Fibroblasts (MEFs)

All animal experiments were approved by the Animal Experimentation Committees of Tokyo University of Science (Y17051, Y18060, Y19056) or the University of Tsukuba (19–274). We back-crossed *Srebp-1c* KO mice on a B6;129S6 background (B6; 129S6-Srebf1tm1Mbr/J; Jackson Laboratory, Bar Harbor, ME, USA) and C57Bl/6J mice (CLEA Japan, Tokyo, Japan) to obtained *Srebp-1c* KO mice on a C57Bl/6 background. All the animals were maintained under specific pathogen-free conditions. At 3 months of age, Wd and *Srebp-1c* KO mice were allocated to two groups: an *ad libitum*-fed (AL) and a CR (70% of the energy intake of AL) group. At 10 months of age, four groups of mice (WdAL, WdCR, KOAL, and KOCR) were provided with food 0.5–1 h prior to turning off the lights in the evening, then they were euthanized under isoflurane anesthesia (Mylan, Canonsburg, PA, USA) 2–3 h later, after which WAT was harvested. The time-course measurement of food intake of the Wd and KO mice that were fed AL, and the body weights of the four groups, are shown in Appendix A.

MEFs were obtained from *Srebp-1c* KO and Wd mice, and *Fgf21* KO and Wd mice on a C57Bl/6 background [36], as previously reported [16]. Briefly, 13–15-day old embryos (E13–15) were collected from pregnant mice of each KO line, minced and trypsinized. MEFs were separated by passing the tryptic digests through a cell strainer.

### 2.2. Cell Culture and Reagent Treatment

3T3-L1 preadipocytes were purchased from the Japanese Collection of Research Bioresources (JCRB) cell bank (Osaka, Japan) and maintained in Dulbecco’s modified Eagle’s medium (DMEM) containing a low glucose concentration (Wako, Osaka, Japan), 10% fetal bovine serum (FBS) (Thermo Fisher Scientific, Waltham, MA USA), and 1% penicillin/streptomycin (P/S) (Sigma-Aldrich, MO, USA). MEFs were maintained in DMEM containing a high glucose concentration (Wako), 10% FBS, 1% P/S, and 0.1 μM 2-mercaptoethanol (Sigma). The differentiation of 3T3-L1 preadipocytes or MEFs to mature adipocytes was achieved by using our previous published protocol [37]. In the present study, 3T3-L1 cells or MEFs were used as mature adipocytes 8–12 or 16 days after the induction of differentiation, respectively. PD1730741 (Funakoshi, Tokyo, Japan) was dissolved in DMSO to make a 5 mM solution, and then it was diluted in PBS. Differentiated 3T3-L1 cells (day 7) were treated with 50 nM PD1730741 for 24 h and then collected.

### 2.3. Retrovirus Plasmid Construction

The construction of the retrovirus plasmids for *Srebp-1c* and *Fgf21* overexpression have been described in our previous reports [16,29]. *Srebp-1a* cDNA was obtained by PCR using KOD FX Neo (Toyobo, Osaka, Japan) and the following primers: 5′-TTT GGA TCC GCC ACC ATG GAC GAG CTG GCC TT-3′ and 5′-TTT GAA TTC TTA CAG GGC CAG GCG GGA-3′. Amplified *Srebp-1a* fragments were digested with BamHI and EcoRI and subcloned into BamHI- and EcoRI-digested pBluescript II SK (+). Then, this plasmid was digested with BamHI and EcoRI, and the gene sequence was inserted into pMXs-AMNN-Puro (pMXs-AMNN-Srebp-1a-Puro) after it was also digested using the same enzymes. The target sequences of the shRNAs against *Lepr* were designed using the Public TRC Portal website (http://www.broadinstitute.org/rnai/public/seq/search), and the sequences were as follows: 5′-GCT AGG TGT AAA CTG GGA CAT CTC GAG ATG TCC CAG TTT ACA CCT AGC TTT TT-3′ and 5′-CGA AAA AGC TAG GTG TAA ACT GGG ACA TCT CGA GAT GTC CCA GTT TAC ACC TAG C-3′. The underlined letters are the sense and antisense target sequences. These oligonucleotides were inserted into BstBI- and PmeI-digested pMXs-puro-mU6 (pMXs-puro-shLeptinR).

### 2.4. Retrovirus Vector Preparation

Retrovirus vectors were generated as reported previously [16]. Briefly, each pMXs-AMNN-Puro plasmid or pMXs-puro-shLeptinR plasmid was transfected into Plat-E cells (kindly provided by T. Kitamura, University of Tokyo, Japan) using the calcium phosphate method. To obtain each overexpressing or shLeptinR-expressing 3T3-L1 cell line, the supernatant from each virus-containing culture was collected after 3 days. The 3T3-L1 cells were infected by incubation in the collected virus-containing supernatant for 2 days, followed by treatment with 2 μg/mL puromycin for a further 5 days. The 3T3-L1 cells overexpressing empty vectors or expressing shRNA targeting GFP (shGFP) were used as control cells.

### 2.5. RT-PCR and Semi-Quantitative RT-PCR

RNA was extracted from WAT and other cell types using ISOGENII (Nippon Gene, Tokyo, Japan). The purified RNA was reverse transcribed using ReverTra Ace^®^ qPCR RT Master Mix (Toyobo) and the cDNAs were then amplified using a CFX Connect™ Real-time System, Thunderbird SYBR qPCR mix, and the primers for each gene. These procedures were performed according to the manufacturer’s protocol. Since the intrinsic expression of *Srebp-1c* mRNA was very low in 3T3-L1 adipocytes, RT-PCR was not possible. Therefore, we performed conventional PCR and agarose gel electrophoresis of the PCR products, followed by ethidium bromide staining. Fluorescence of the ethidium bromide was visualized using an LAS3000 (Fujifilm, Tokyo, Japan) and data were analyzed using Multigauge software (Fujifilm). Target gene expression data were normalized to *Rps18* expression (*n* = 4). The primer pair sequences are shown in Table 1.

### 2.6. Western Blotting

Cell lysis and immunoblotting were performed as previously described [37]. Briefly, the collected cells were lysed in lysis buffer (50 mM Tris-HCl (pH 6.8), 2% SDS, 3 M urea, 6% glycerol), boiled for 5 min, and sonicated. Lysates containing 15 μg protein were subjected to SDS/PAGE and the proteins were then transferred to nitrocellulose membranes. The membranes were blocked with 2.5% skim milk and 0.25% bovine serum albumin in Tris-buffered saline (50 mM Tris-HCl (pH 7.4) and 150 mM NaCl) containing 0.1% Tween 20 (TTBS) for 60 min at room temperature, then incubated with appropriate primary antibodies overnight at 4 °C. Primary antibodies against FGF21 (Abcam, Cambridge, UK), PGC-1α (Sigma, MO, USA AB3242), mitochondrial transcription factor A (TFAM) (Proteintech, Chicago, IL, USA, 19998-1-AP), SIRT3 (cell signaling technology (CST), Beverly, MA, USA, #5490), ACC (CST, #3662), p-STAT3 (Thermo Fisher Scientific, 44-3804), and LaminB1 (Medical & Biological Laboratories, Nagoya, Japan, PM064) were used. The membranes were then incubated with an appropriate secondary antibody (a horseradish peroxidase-conjugated F(ab’)2 fragment of goat anti-mouse IgG or anti-rabbit IgG; Jackson Immuno Research, West Grove, PA, USA) for 60 min at room temperature. Thereafter, they were incubated with ImmunoStar LD (Wako), specific protein bands were visualized using an LAS3000 (Fujifilm, Tokyo, Japan), and the data were analyzed using Multigauge software (Fujifilm).

### 2.7. Statistical Analysis

The values presented are means ± standard deviations (SDs). The data were statistically evaluated using Student’s *t*-test, two-way ANOVA and/or Tukey’s test, with R software (Version 3.4.1, R Foundation for Statistical Computing, Vienna, Austria). *p* < 0.05 was considered to represent statistical significance.

## 3. Results

### 3.1. Role of SREBP-1c in the Effects of CR on Gene Expression in WAT

We have reported previously that CR increases the expression of *Srebp-1c*, *Srebp-1a*, and *Pgc-1a* mRNAs in the WAT of Wd B6;129S6 mice but not in KO mice [16]. In the WAT of mice on a C57Bl/6 background, CR also increased the expression of *Srebp-1c*, *Pgc-1a*, and *Fgf21* mRNAs in Wd mice but not in KO mice (Figure 1A or Figure 1C,D). Similar findings about the CR-associated upregulation of *Fgf21* mRNA were observed in B6;129S6 mice (Appendix A). However, in contrast to mice on a B6;129S6 background, CR did not upregulate the expression of *Srebp-1a* mRNA in either Wd or KO mice on a C57Bl/6 background (Figure 1B). Overall, it is likely that the CR-associated upregulation of these factors is less exaggerated in C57bl/6 mice compared with B6;129S6 mice.

### 3.2. Effects of SREBP-1c on the Expression of Genes and Proteins Involved in FA Biosynthesis and Mitochondrial Biogenesis

To confirm that SREBP-1c is the significant regulator of the expression of genes and proteins involved in FA biosynthesis and mitochondrial biogenesis in vitro, we generated 3T3-L1 preadipocytes that overexpressed SREBP-1c (SREBP-1c OE) using retroviral vectors. The SREBP-1c OE 3T3-L1 preadipocytes were differentiated to adipocytes (Figure 2A), and the expression levels of mRNAs and proteins of interest were analyzed.

In SREBP-1c OE adipocytes, the expression of *Srebp-1a* mRNA was similar to that of control cells (Figure 2B), whereas that of *perilipin A* (*PeriA*) and *adiponectin* (*Adipoq*), which are markers of adipocyte differentiation, was upregulated (Figure 2C,D). Fatty acid synthase (FASN) is a rate-limiting enzyme in FA biosynthesis. The expression of *Fasn* mRNA was high in SREBP-1c OE adipocytes (Figure 2E). Moreover, the expression of both *Fgf21* and *Pgc-1a* mRNAs was high in OE adipocytes (Figure 2F,G). In addition, the protein expression of acetyl-CoA carboxylase (ACC) and malic enzyme 1 (ME1), which are FA biosynthetic enzymes, and FGF21 and PGC-1α were high in OE adipocytes (Figure 2H–L). With regard to mitochondrial proteins, expression of SIRT3 was also high in SREBP-1c OE adipocytes, but that of TFAM was not (Figure 2H or Figure 2M,N).

To further characterize the regulation of FA biosynthetic genes by FGF21 and PGC-1α, we generated Wd and KO adipocytes by differentiating MEFs derived from Wd and *Srebp-1c* KO mice. The expression of *Srebp-1c* was not detectable in KO adipocytes and that of *Srebp-1a* was similar to that of control adipocytes (Figure 3A,B). The expression levels of *PeriA* and *Adipoq* mRNAs in KO adipocytes did not differ from those in Wd adipocytes, suggesting that SREBP-1c deficiency did not alter differentiation (Figure 3C,D). However, the expression of *Fasn* was lower in KO adipocytes than Wd adipocytes (Figure 3E). The expression of *Fgf21* mRNA was slightly reduced, while that of *Pgc-1a* mRNA was significantly lower in KO adipocytes (Figure 3F,G). Taken together, our findings suggest that the expression of both *Pgc-1a* and *Fgf21* is positively regulated by Srebp-1c, in addition to that of *Fasn*. Moreover, SREBP-1c is the significant regulator of the expression of genes and proteins involved in FA biosynthesis and mitochondrial biogenesis in adipocytes.

### 3.3. Roles of FGF21 and PGC-1α in Mitochondrial Biogenesis

CR upregulated the expression of both *Fgf21* and *Pgc-1α* mRNAs and proteins via SREBP-1c. Therefore, we next determined the roles of FGF21 and PGC-1α in mitochondrial biogenesis.

In FGF21 OE adipocytes (Figure 4A or Figure 4C,D), the expression of *Pgc-1α* mRNA and PGC-1α protein was high (Figure 4B,C or Figure 4E,F). Treatment with PD173074, an FGF receptor (FGFR) inhibitor, reduced the phosphorylation of ERK without reducing *Fgf21* mRNA and FGF21 protein expression (Figure 4A or Figure 4C–E). In addition, this treatment did not reduce the expression of *Pgc-1α* mRNA or PGC-1α protein (Figure 4B,C or Figure 4F).

In Fgf21 KO adipocytes differentiated from MEFs, the expression levels of *PeriA*, *Adipoq*, and *Pgc-1α* mRNAs were much lower than in control cells, suggesting that the significant reduction in *Pgc-1α* mRNA expression is associated with impaired adipocyte differentiation (Figure 5A–D). 

Taken together, these findings indicate that FGF21 positively regulates PGC-1*α* expression, but ERK signaling does not have a significant effect.

### 3.4. Effect of Leptin Signaling on the Expression of Genes and Proteins Involved in FA Biosynthesis and Mitochondrial Biogenesis

To determine the effect of leptin signaling on FA biosynthesis and mitochondrial biogenesis in adipocytes, we knocked down leptin receptor expression in 3T3-L1 preadipocytes using a retroviral vector and analyzed the cells after differentiating them to adipocytes (Figure 6A). The activation of the leptin receptor phosphorylates STAT3, the major downstream target molecule of leptin signaling [38]. Leptin receptor knockdown (KD) reduced the phosphorylation of STAT3, confirming that leptin signaling had been inhibited (Figure 6F,G). The expression of *Srebp-1c*, *Srebp-1a*, *Fgf21*, and *Pgc-1α* mRNAs was significantly higher in leptin receptor KD adipocytes than in control cells (Figure 6B–E). Moreover, the expression of FGF21, PGC-1α, TFAM, ACC, and ME-1 proteins was higher in leptin receptor KD adipocytes (Figure 6F or Figure 6H–L). This finding suggests that a reduction in leptin signaling induces the expression of both *Srebp-1c* and *Srebp-1a* mRNAs and the expression of proteins involved in FA biosynthesis and mitochondrial biogenesis in adipocytes.

## 4. Discussion

SREBP-1 is a family of transcription factors that are master regulators of FA biosynthesis. It comprises two isoforms, SREBP-1c and SREBP-1a. SREBP-1c is upregulated in the livers of obese mice [39,40], and fatty liver occurs in mice with liver-specific overexpression of SREBP-1c [41]. According to previous findings, SREBP-1c functions predominantly in the liver, rather than in WAT, and is involved in hepatic steatosis [42,43]. However, we have reported previously that CR upregulates the expression of genes and/or proteins involved in FA biosynthesis and mitochondrial biogenesis, including *Pgc-1a* mRNA expression via SREBP-1c, in WAT rather than in the livers of mice on a B6;129S6 background [16]. In the present study, we have shown similar results in mice on a C57Bl/6 background and mature adipocytes differentiated from MEFs. We also demonstrated that CR upregulates FGF21 via SREBP-1c in mice on both B6;129S6 and C57Bl/6 backgrounds. *Fgf21* and *Pgc-1a* mRNA transcripts were downregulated in adipocytes differentiated from MEFs derived from SREBP1c KO mice, but these gene expressions were unchanged in WAT between WdAL and KOAL mice. We are not able to explain the discrepancy between the in vitro and in vivo findings, but these results reveal, at least, that SREBP-1c positively regulates both gene expressions. We also generated SREBP-1a OE 3T3-L1 adipocytes (Appendix AA) and the expression levels of mRNAs and proteins of interest were analyzed. Since the intrinsic expression of *Srebp-1c* mRNA was very low in 3T3-L1 adipocytes, RT-PCR was not possible. In SREBP-1a OE adipocytes, the expression of *Srebp-1c* mRNA was similar in control and SREBP-1a OE adipocytes (Appendix AB). However, the expression of *PeriA* and *Adipoq* mRNA was high in OE cells (Appendix AC,D), as was that of *Fasn* and *Fgf21* mRNAs, but this was not the case for *Pgc-1a* mRNA (Appendix AE–G). In addition, the protein expression of ACC, ME-1, FGF21, PGC-1α, and SIRT3 was unaffected or downregulated by OE (Appendix AH–N). In contrast, protein expression of TFAM was high in SREBP-1a OE adipocytes but not in SREBP-1c adipocytes. Moreover, *Fgf 21* mRNA expression was upregulated in both SREBP-1c and -1a OE 3T3-L1 adipocytes. In contrast, FGF21 protein was increased in SREBP-1c OE adipocytes but decreased in SREBP-1a OE adipocytes. We are not able to rationally explain these distorted results. However, overall, by comparing SREBP-1c and SREBP-1a OE 3T3-L1 adipocytes, we have confirmed that SREBP-1c, rather than SREBP-1a, is principally responsible for increases in the expression of genes and proteins involved in FA biosynthesis in mature adipocytes.

It has been reported that CR activates mitochondrial biogenesis in various tissues, including WAT, liver, heart, and skeletal muscle [44,45]. However, we found that CR-induced mitochondrial biogenesis is mediated by SREBP-1c only in WAT [16]. ME-1, which is upregulated in SREBP-1c OE adipocytes, is one of the enzymes involved in the pyruvate/malate cycle. Our previous proteomic analysis showed that CR upregulates the expression of proteins involved in the pyruvate/malate cycle, including ATP-citrate lyase, citrate synthase, mitochondrial pyruvate dehydrogenase E1 component subunit beta, and mitochondrial pyruvate carboxylase, as well as ME-1. Therefore, we hypothesized that CR might activate the pyruvate/malate cycle in WAT in order to switch from the use of glucose to the use of energy-dense FAs so that energy can be used more efficiently under poor food supply conditions [46]. On the basis of this hypothesis, it makes sense that CR would simultaneously upregulate the expression of proteins involved in FA biosynthesis, the pyruvate/malate cycle, and mitochondrial biogenesis via SREBP-1c. Bruss et al. has shown that CR activates de novo FA biosynthesis predominantly in WAT, rather than in the liver [47], and their findings are consistent with this hypothesis.

FGF21 positively regulates PGC-1α and PPARγ via feed-forward autocrine/paracrine loops in WAT [26,27]. It is widely accepted that PGC-1α is a master regulator of CR-associated mitochondrial biogenesis [18]. We have shown here that CR upregulates the expression of both FGF21 and PGC-1α via SREBP-1c. CR also upregulates PPARγ expression in WAT [29]. Therefore, we characterized the reciprocal regulatory mechanism involving FGF21 and PGC-1α expression in mitochondrial biogenesis. In *Fgf21* KO adipocytes, the expression of PGC-1α was low and adipocyte differentiation was impaired. However, in FGF21 OE adipocytes, the expression of *Pgc-1a* mRNA and PGF-1α protein was high [29]. FGF21 promotes the phosphorylation of ERK via the binding of FGF21 to FGFR and the β-klotho (KLB) receptor complex [21]. Treatment with an FGFR inhibitor reduced the phosphorylation of ERK and the expression of *Pgc-1a* mRNA but not that of PGC-1α protein. In SREBP-1c OE adipocytes, the expression of FGF21, PGC-1α, and SIRT3 proteins was very high. We previously demonstrated that SREBP-1c binds to the promoter of the *Pgc-1a* gene in adipocytes derived from Wd MEFs, but this did not occur in SREBP-1c KO MEFs [16]. Furthermore, in brown adipocytes, it has been reported that SREBP-1c activates the *Pgc-1a* promoter [48]. Therefore, when CR induces PGC-1α expression in adipocytes, it is likely that direct transcriptional regulation by SREBP-1c is more significant than the induction of FGF21 by SREBP-1c.

Leptin is secreted by WAT and acts as a satiety signal to the hypothalamus, activating NPY and AGRP neurons and suppressing POMC and CART neurons in the hypothalamus, subsequently activating the sympathetic nervous system and thereby lipolysis in WAT via β3-adrenergic receptors [32]. We have shown that a reduction in leptin signaling increases the expression of SREBP-1c, SREBP-1a, SREBP-1-regulated genes, FGF 21, and PGC-1α in mature adipocytes. These findings suggest that lower leptin secretion reduces leptin receptor signaling via an autocrine/paracrine loop, resulting in the greater expression of genes involved in FA biosynthesis and mitochondrial biogenesis in the WAT of CR mice. Previously, we found that the expression of proteins involved in FA biosynthesis is higher in obese fa/fa Zucker rats that have a leptin receptor mutation than in lean +/+ rats. Moreover, CR increases the expression of proteins involved in FA biosynthesis in lean +/+ rats but not in obese fa/fa Zucker rats [49]. Our present in vitro findings are consistent with these findings in Zucker rats.

Based on findings concerning the CR-associated metabolic remodeling of WAT in *Srebp-1c* KO mice, we investigated the upstream and downstream regulatory mechanisms of SREBP-1c in vitro. To confirm our results in vitro, we examined the mRNA and protein levels of most factors in both OE cells and KD or KO cells, and we were able to obtain relatively consistent data with regard to upregulated and downregulated genes. As a result, it was likely that a reduction in leptin signaling induced the expression of proteins involved in FA biosynthesis and mitochondrial biogenesis via SREBP-1c in adipocytes. PGC-1α is upregulated via both the direct transcriptional regulation of SREBP-1c and the upregulation of FGF21 indirectly regulated by SREBP-1c, but it is unlikely that the FGF21-associated upregulation of PGC-1α expression is a predominant factor in mitochondrial biogenesis induced by SREBP-1c. Therefore, we conclude that CR might downregulate an autocrine/paracrine loop involving leptin, with a reduction in leptin signaling activating de novo FA biosynthesis and mitochondrial biogenesis through the upregulation of SREBP-1c in WAT, in addition to the effects that leptin exerts via the central nervous system. SREBP-1c expression is high when the leptin concentration is low, and this is regulated in a GH/IGF-1-independent manner, but it is a key player in the CR-associated metabolic remodeling of WAT, which involves the upregulation of both FA biosynthesis and mitochondrial biogenesis. The CR-associated metabolic remodeling of WAT might be a leptin-mediated adaptive response to food shortage, causing a switch from the use of glucose to lipid as an energy substrate.

## Figures and Tables

**Figure 1 nutrients-12-02054-f001:**
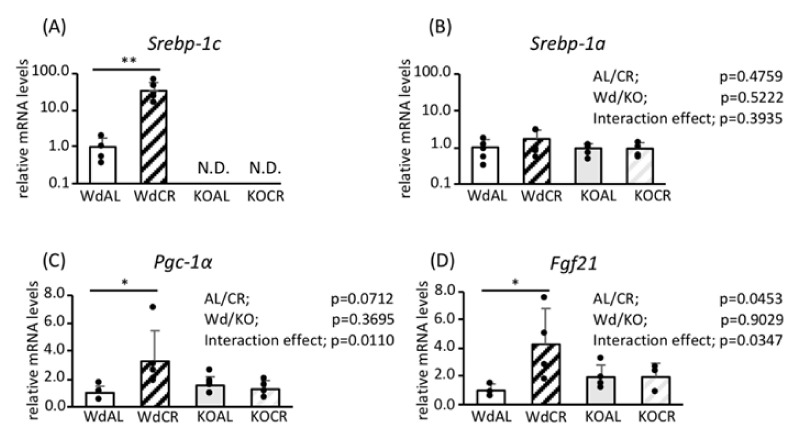
The effects of *Srebp-1c* KO on the expression of key regulators of CR-associated metabolic remodeling in the WAT of mice on a C57Bl/6 background. The mRNA expression levels of *Srebp-1c* (**A**), *Srebp-1a* (**B)**, *Fgf21* (**C**), and *Pgc-1a* (**D**) in WAT were measured using RT-PCR and were normalized to *Tbp* expression (*n* = 4). Values are means ± SDs. * *p* < 0.05, ** *p* < 0.01 vs. AL, according to Student’s *t*-test, or two-way ANOVA and Tukey’s test.

**Figure 2 nutrients-12-02054-f002:**
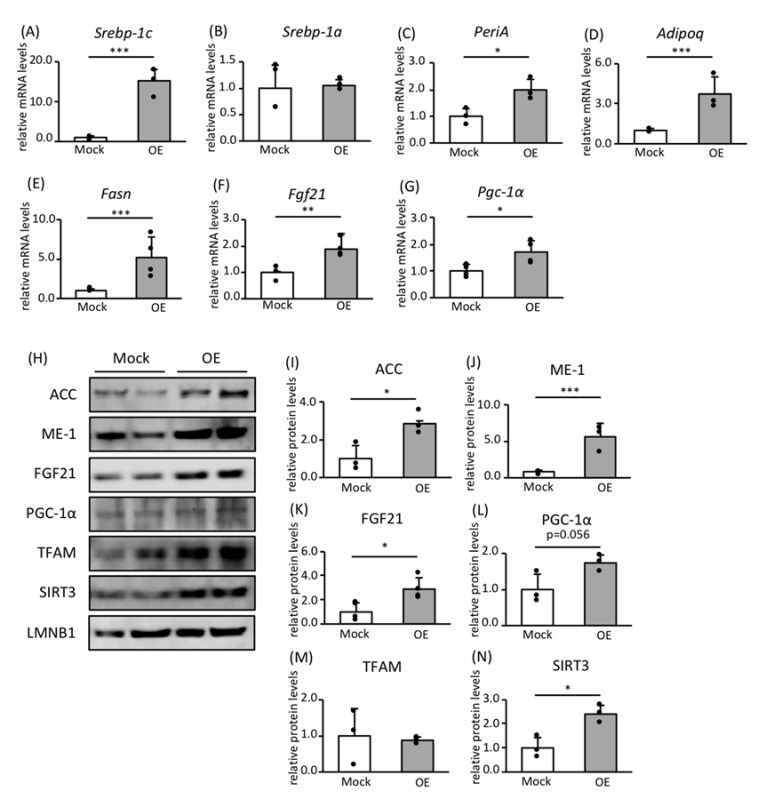
The effects of SREBP-1c overexpression on the expression of genes and proteins involved in fatty acid (FA) biosynthesis and mitochondrial biogenesis in mature 3T3-L1 adipocytes. Control and SREBP-1c OE preadipocytes were differentiated into mature adipocytes in four separate dishes from each phenotype. RNA was extracted and lysates were prepared from each dish. RNA was extracted and lysates were prepared from adipocytes. The mRNA expression levels of *Srebp-1c* (**A**), *Srebp-1a* (**B**), *PeriA* (**C**), *Adipoq* (**D**), *Fasn* (**E**), *Fgf21* (**F**), and *Pgc-1a* (**G**) were determined using RT-PCR and normalized to *Rps18* expression (*n* = 4). (**H**) Representative immunoblot images, showing the expression levels of proteins involved in FA biosynthesis and mitochondrial biogenesis. Quantitative analysis was performed using a chemiluminescence method. The protein expression of ACC (**I**), ME-1 (**J**), FGF21 (**K**), PGC-1α (**L**), TFAM (**M**), and SIRT3 (**N**) are shown as the relative intensities of the indicated protein divided by that of LMNB1 as an internal control (*n* = 4). Values are means ± SDs. * *p* < 0.05, ** *p* < 0.01, *** *p* < 0.001 vs. controls, according to Student’s *t*-test.

**Figure 3 nutrients-12-02054-f003:**
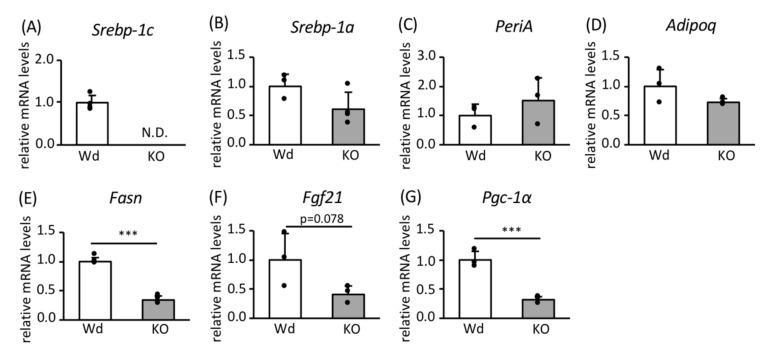
The effects of SREBP-1c deficiency on the expression of adipocyte differentiation markers and genes involved in FA biosynthesis and mitochondrial biogenesis in mature adipocytes. MEFs were obtained from four individual embryos of either Wd or *Srebp-1c* KO mice, differentiated to mature adipocytes, and then RNA was extracted from each dish. The mRNA expression levels of *Srebp-1c* (**A**), *Srebp-1a* (**B**), *PeriA* (**C**), *Adipoq* (**D**), *Fasn* (**E**), *Fgf21* (**F**), and *Pgc-1a* (**G**) were determined using RT-PCR and normalized to *Rps18* expression (*n* = 4). Values are means ± SDs. *** *p* < 0.001 vs. Wd, according to Student’s *t*-test.

**Figure 4 nutrients-12-02054-f004:**
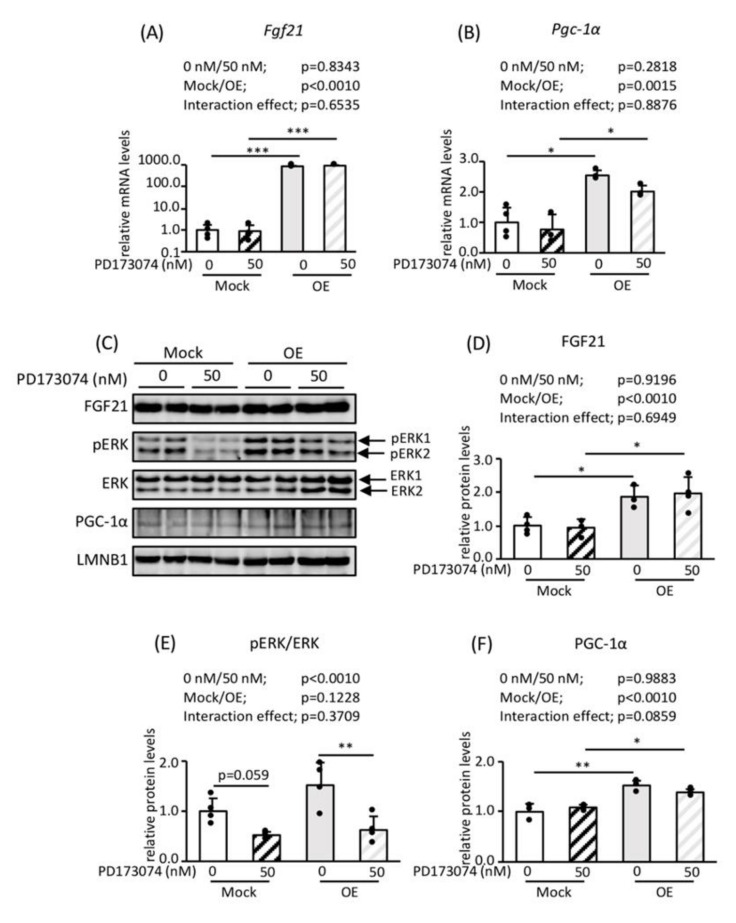
The effects of FGF21 overexpression and the inhibition of FGFR on the expression of genes and proteins involved in FGF21 signaling and PGC-1α in mature 3T3-L1 adipocytes. Control and FGF21 OE preadipocytes were differentiated into mature adipocytes in four separate dishes for each phenotype, and they were treated with or without 50 nM PD173074, an FGFR inhibitor, for 24 h. (**A**,**B**) RNA was extracted and lysates were prepared from each dish. The mRNA expression levels of *Fgf21* (**A**) and *Pgc-1a* (**B**) were determined using RT-PCR and normalized to *Rps18* expression (*n* = 4). (**C**) Representative immunoblot images showing the expression of proteins involved in FGF21 signaling and mitochondrial biogenesis. Quantitative analysis was performed using a chemiluminescence method. The protein expression of FGF21 (**D**) and PGC-1α (**F**) is shown as the relative intensity of the indicated protein divided by that of LMNB1 as an internal control (*n* = 4). Extracellular signal-regulated kinase (ERK) phosphorylation is expressed as the relative intensity of the phosphorylated form of ERK/total ERK (*n* = 4) (**E**). Values are means ± SDs **p* < 0.05, ***p* < 0.01, ****p* < 0.001 vs. controls administered the same treatment.

**Figure 5 nutrients-12-02054-f005:**
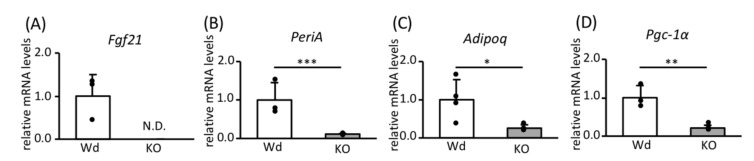
The effects of FGF21 deficiency on the expression of adipocyte differentiation markers and genes involved in mitochondrial biogenesis in mature adipocytes. MEFs were obtained from four individual embryos of either Wd or *Fgf21* KO mice, differentiated to mature adipocytes, and then RNA was extracted from each dish. The mRNA expression levels of *Fgf21* (**A**), *PeriA* (**B**), *Adipoq* (**C**), and *Pgc-1a* (**D**) were analyzed using RT-PCR and normalized to *Rps18* expression (*n* = 4). Values are means ± SDs. * *p* < 0.05, ** *p* < 0.01 vs. Wd, according to Student’s *t*-test, ****p* < 0.001.

**Figure 6 nutrients-12-02054-f006:**
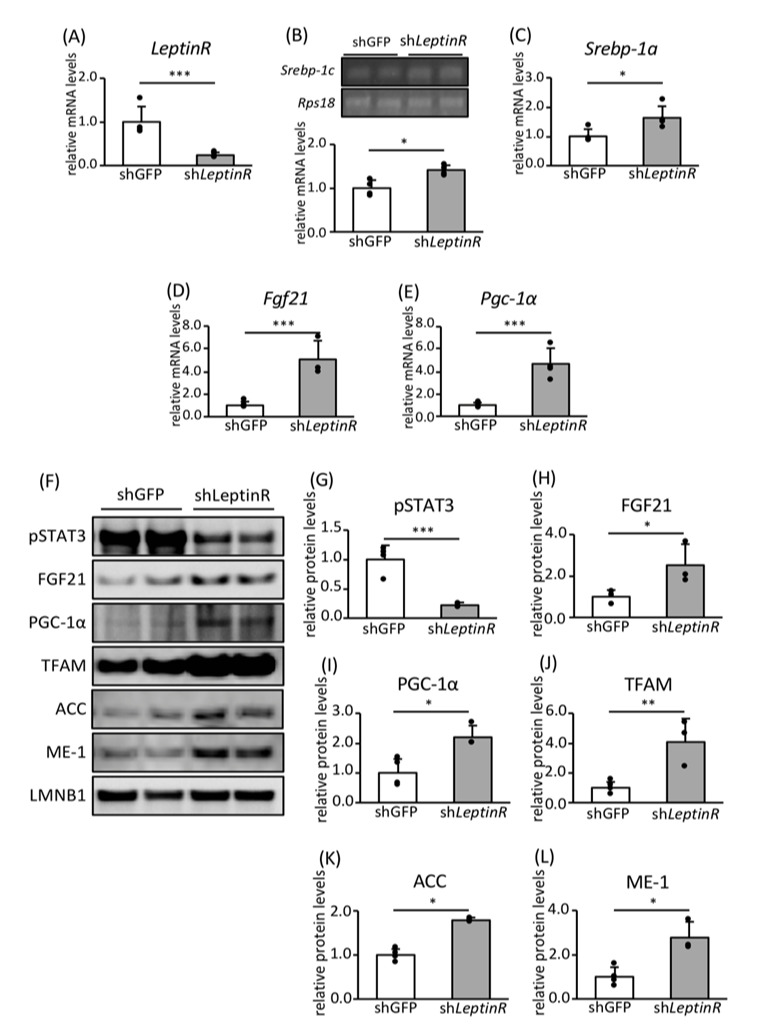
The effects of leptin signaling on the expression of genes and proteins involved in leptin signaling, FA biosynthesis, and mitochondrial biogenesis in mature 3T3-L1 adipocytes. LeptinR knockdown (KD) (*shLeptinR*) and control (*shGFP*) preadipocytes were differentiated into mature adipocytes in four separate dishes for each phenotype, and then RNA was extracted, and lysates were prepared from each dish. The mRNA expression levels of *LeptinR* (**A**), *Srebp-1a* (**C**), *Fgf21* (**D**), and *Pgc-1a* (**E**) were determined using RT-PCR and normalized to *Rps18* expression (*n* = 4). (**B**) Representative images of ethidium bromide-stained gels, showing fluorescence corresponding to the products of *Srebp-1c* cDNA amplification by RT-PCR. Semiquantitative analysis was performed and the data were normalized to *Rps18* expression (n = 4). (**F**) Representative immunoblot images showing the expression of proteins involved in leptin signaling, FA biosynthesis, and mitochondrial biogenesis. Quantitative analysis was performed using a chemiluminescence method. The protein expression of pSTAT (**G**), FGF21 (**H**), PGC-1α (**I**), TFAM (**J**), ACC (**K**), and ME-1 (**L**) is shown as the relative intensity of the indicated protein divided by that of LMNB1 as an internal control (*n* = 4). Values are means ± SDs. * *p* < 0.05, ** *p* < 0.01, *** *p* < 0.001 vs. shGFP, according to Student’s *t*-test.

**Table 1 nutrients-12-02054-t001:** List of primers for RT-PCR.

	Forward	Reverse
Adipoq	5′-TGC CGA AGA TGA CGT TAC AAC-3′	5′-CTT CAG CTC CTG TCA TTC CAA C-3′
Fasn	5′-AGC AGG CAC ACA CAA TGG AC-3′	5′-GAA GAA AGA GAG CCG GTT G-3′
Fgf21	5′-GAA GCC CAC CTG GAG ATC AG-3′	5′-CAA AGT GAG GCG ATC CAT AGA G-3′
LeptinR	5′- CAG TCT TCGG GGA TGT GAA TG-3′	5′- CAT TGT TTG GCT GTC CCA AG-3′
PeriA	5′-TGG GAA GCA TCG AGA AGG TG-3′	5′-ATG GTG TGT CGA GAA AGA GTG TTG-3′
Pgc-1α	5′-AGA CGG ATT GCC CTC ATT TG-3′	5′-CAG GGT TTG TTC TGA TCC TGT G-3′
Rps18	5′-TGC GAG TAC TCA ACA CCA ACA T-3′	5′-CTT TCC TCA ACA CCA CAT GAG C-3′
Srebp-1a	5′-GGC CGA GAT GTG CGA ACT-3′	5′-TTG TTG ATG AGC TGG AGC ATG T-3′
Srebp-1c	5′-GGA GCC ATG GAT TGC ACA TT-3′	5′-GGC CCG GGA AGT CAC TGT-3′
Tbp	5′-CCC TCA CAC TCA GAT CAT CTT CTC-3′	5′-GCC TTG TCC CTT GAA GAG AAC C-3′

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
