# Peer review of "Srebp-1c/Fgf21/Pgc-1α Axis Regulated by Leptin Signaling in Adipocytes—Possible Mechanism of Caloric Restriction-Associated Metabolic Remodeling of White Adipose Tissue"

_nutrients, 2020, doi:10.3390/nu12072054_

Round 1

Reviewer 1 Report

The authors have taken into consideration the comments given in the first revision and the answers to the different concerns raised are satisfactorily addressed. The present version is clearer and rigorous. The focus of the manuscript has been appropriately improved. This reviewer thanks the authors for the answers given.

Author Response

We appreciate the Reviewer #1's positive evaluation.

Reviewer 2 Report

General comments:

The authors assessed the role of SREBP1c, FGF21, PGC1a and leptin signalling in adipose tissue in response to caloric restriction in mice and in cell lines. The main finding is that the upregulation SREBP1c, FGF21, PGC1a by calorie restriction is lost in SREBP1c KO mice, and that overexpression of these regulatory molecules in adipocyte cell lines reveals that FGF21 and leptin signalling maybe important for SREBP1c induced stimulation of PGC1a. The manuscript provides considerable amount of information. However, the relevance of a majority of the in vitro findings to caloric restriction, the novelty of the in vivo findings, and the rationale for the use of cell lines without nutrient restriction/deprivation, are not clear to me. There are major issues that need to be addressed.

Major comments:

Lines 199-206 and Fig 1: The authors previously reported that caloric restriction upregulates SREBP1c, and relevant lipogenic and mitochondrial biogenic markers, in adipose tissue of wild type but not SREBP1c KO mice on a B6;129S6 background (PMID 28256090). The incremental knowledge gained from confirming similar findings in C57Bl/6J is unclear. Also, are any of the other markers (eg. PeriA, AdipoQ, FASN, ACC, TFAM, SIRT3) that were studied in cell lines altered by caloric restriction in wild type and KO mice? Also, at a minimum, data on the actual caloric intake and body weight of the mice needs to be provided as the focus of this paper is on caloric restriction.

Lines 212-228, 289-293, Figs 2, 3 and 5: The rationale for overexpressing both SREBP1c, SREBP1a, and FGF21 in 3T3-L1 adipocytes is not well justified. Though you have shown that overexpression of SREBP1c or FGF21 does upregulate FASN, ACC, FGF21, PGC1a, or other markers, it is not clear to me how the upregulation of these markers is causative to the adipose adaptations to caloric restriction. Please provide data showing that nutrient restriction/deprivation upregulates SREBP1c, FGF21, and other pertinent markers in these cell lines. In the absence of such information, the findings as presented may simply imply an independent regulatory role for SREBP1c and FGF21 on these markers and does not necessarily mean that these markers are involved in caloric restriction. Further, the overexpression of FGF21 upregulated FGF21 by ~1000 fold in cell lines (Fig 5A), whereas, caloric restriction upregulated FGF21 by ~4-fold (Fig 1D). The physiological relevance of these in vitro data needs to be better reconciled.

Lines 244-249 and Fig 3. If SREBP1a doesn’t change with caloric restriction (Fig 1B), then it doesn’t make any sense to me on why the effects of it’s overexpression are being studied (Fig 3). Also, SREBP1a overexpression upregulates FGF21 mRNA (Fig 3F) but downregulates it’s protein (Fig 3K) – there is no discussion on this apparent discrepancy.

Lines 264-271 and Fig 4: The rationale for generating adipocytes from SREBP1c KO mice is not well justified and some of the data contradict data in Fig 1. For example, FASN, FGF21 and PGC1a transcripts are downregulated in adipocytes generated from SREBP1c KO mice (Fig 4 4E-G), but PGC1 and FGF21 are unaltered between wild type and KO mice (Fig 1 C, D). There is no explanation for this apparent discrepancy.

Lines 320-330 and Fig 7: It’s interesting that silencing of leptin receptor signalling upregulates FGF21, PGC1a, SREBP1c and 1a. As SREBP regulates leptin, is the expression of leptin transcripts and protein impacted by overexpression of SREBP1c and FGF21, or by knockdown of leptin receptor in 3T3 cells?

Lines 36-38, 368-370, 393-395, 409-417: At multiple instances throughout the manuscript the authors imply causation from data generated from cell lines to the in vivo caloric restriction model. But there is no evidence provided to support such claims. Also, majority of the introduction and discussion is overly long with extraneous material which can be condensed.  

Author Response

Reviewer's comment

Lines 199-206 and Fig 1: The authors previously reported that caloric restriction upregulates SREBP1c, and relevant lipogenic and mitochondrial biogenic markers, in adipose tissue of wild type but not SREBP1c KO mice on a B6;129S6 background (PMID 28256090). The incremental knowledge gained from confirming similar findings in C57Bl/6J is unclear. Also, are any of the other markers (eg. PeriA, AdipoQ, FASN, ACC, TFAM, SIRT3) that were studied in cell lines altered by caloric restriction in wild type and KO mice? Also, at a minimum, data on the actual caloric intake and body weight of the mice needs to be provided as the focus of this paper is on caloric restriction.

Our response

In our previous paper (PMID: 28256090), we showed that CR upregulates Pgc-1α expression in an Srebp-1c dependent manner in mice on a B6;129S6 background, but did not show an Srebp-1c dependency in the CR-associated upregulation of Fgf21 in mice on a B6;129S6 background. Therefore, we showed here first that CR increases Fgf21 expression in an Srebp-1c dependent manner in mice on a C57Bl/6J background as well as a B6;129S6 background. Since we have already shown the other markers (eg. PeriA, AdipoQ, FASN, ACC, TFAM, SIRT3) previously, we checked the data, but have not shown here.

As the reviewer recommended, we show the data of food intake and body weight of the mice as Supplementary Figure S1.

Reviewer's comment 

Lines 212-228, 289-293, Figs 2, 3 and 5: The rationale for overexpressing both SREBP1c, SREBP1a, and FGF21 in 3T3-L1 adipocytes is not well justified. Though you have shown that overexpression of SREBP1c or FGF21 does upregulate FASN, ACC, FGF21, PGC1a, or other markers, it is not clear to me how the upregulation of these markers is causative to the adipose adaptations to caloric restriction. Please provide data showing that nutrient restriction/deprivation upregulates SREBP1c, FGF21, and other pertinent markers in these cell lines. In the absence of such information, the findings as presented may simply imply an independent regulatory role for SREBP1c and FGF21 on these markers and does not necessarily mean that these markers are involved in caloric restriction. Further, the overexpression of FGF21 upregulated FGF21 by ~1000 fold in cell lines (Fig 5A), whereas, caloric restriction upregulated FGF21 by ~4-fold (Fig 1D). The physiological relevance of these in vitro data needs to be better reconciled.

Our response

In vitro model of CR is not established so far. In general, it is well known that nutrient restriction/deprivation in vitro is not able to mimics CR in vivo. In this report, therefore, we wanted to show simply that the overexpression of Srebp-1c or Fgf21 induces the upregulation of various CR-associated biomarkers.

As the reviewer pointed out, the overexpression of FGF21 upregulated FGF21 by ~1000 fold in cell lines (Fig 5A), whereas, caloric restriction upregulated FGF21 by ~4-fold (Fig 1D) in mRNA level. We cannot explain the rational reason about the difference in increased degree between mRNA and protein expression. However, since FGF21 protein was upregulated by less than 2-fold in vitro, we think that our findings could reflect the physiological condition.

Reviewer's comment

Lines 244-249 and Fig 3. If SREBP1a doesn’t change with caloric restriction (Fig 1B), then it doesn’t make any sense to me on why the effects of it’s overexpression are being studied (Fig 3). Also, SREBP1a overexpression upregulates FGF21 mRNA (Fig 3F) but downregulates it’s protein (Fig 3K) – there is no discussion on this apparent discrepancy.

Our response

We previously found the CR-associated upregulation of Srebp-1a in mice on a B6;129S6 background. Therefore, we performed the SREBP-1a OE experiment as well as Srebp-1c OE experiment. As reviewer pointed out, however, we agree with the reviewer’s comment that Figure 3 is not important in this manuscript. We moved the Figure 3 to supplementary Figure S3. We just commented in the first paragraph in Discussion section.

Reviewer's comment

Lines 264-271 and Fig 4: The rationale for generating adipocytes from SREBP1c KO mice is not well justified and some of the data contradict data in Fig 1. For example, FASN, FGF21 and PGC1a transcripts are downregulated in adipocytes generated from SREBP1c KO mice (Fig 4 4E-G), but PGC1 and FGF21 are unaltered between wild type and KO mice (Fig 1 C, D). There is no explanation for this apparent discrepancy.

Our response

We appreciate the reviewer’s comment. Figure 4 just showed that the Srebp-1c positively regulates both Pgc-1α and Fgf21 transcripts as well as Fasn. We are not able to rationally explain about the discrepancy between Fig.1 and Fig. 4, but comment in the first paragraph in Discussion section.  

Reviewer's comment

Lines 320-330 and Fig 7: It’s interesting that silencing of leptin receptor signalling upregulates FGF21, PGC1a, SREBP1c and 1a. As SREBP regulates leptin, is the expression of leptin transcripts and protein impacted by overexpression of SREBP1c and FGF21, or by knockdown of leptin receptor in 3T3 cells?

Our response

We did not examine the leptin expression in either 3T3-L1 adipocytes overexpressing SREBP1c and FGF21, or those knocking down leptin receptor.

Reviewer's comment

Lines 36-38, 368-370, 393-395, 409-417: At multiple instances throughout the manuscript the authors imply causation from data generated from cell lines to the in vivo caloric restriction model. But there is no evidence provided to support such claims. Also, majority of the introduction and discussion is overly long with extraneous material which can be condensed.

Our response

In general, since in vitro experiments perform to simplify in vivo models, the findings derived from in vitro experiments support in vivo data. Based on this concept, we performed these experiments described in our manuscript. Therefore, we are not able to agree with the reviewer’s criticism.

To shorten the Introduction section, we rewrote the section, however, we were not able to shorten Discussion section because the additional comments were required to response to suggestion and criticism raised by reviewer.

Reviewer 3 Report

The work presented by Masaki Kobayashi and colleagues showed that Srebp-1c/Fgf21/Pgc-1α axis regulated by leptin signaling in adipocytes which is a possible mechanism of caloric restriction-associated metabolic remodeling of white adipose tissue. This study is novel, and the manuscript is well written. However, I just have some minor concerns.

  • In Figure 2H, the expression of the internal control (LMNB1) is not nearly the same in all the wells, the quantification of Figure 2L and 2M is not accurate based on the western blots’ images in Figure 2H. Similarly, the quantification of Figure 3L and 3M is not seeming to be right based on the western blots’ images in Figure 3H. Same way please also check the quantification in Figure 5F.
  • It will be better if the authors can add another Figure showing the overall model of their findings.

Author Response

Reviewer's comment

In Figure 2H, the expression of the internal control (LMNB1) is not nearly the same in all the wells, the quantification of Figure 2L and 2M is not accurate based on the western blots’ images in Figure 2H. Similarly, the quantification of Figure 3L and 3M is not seeming to be right based on the western blots’ images in Figure 3H. Same way please also check the quantification in Figure 5F.

Our response

We appreciate the positive evaluation and the reviewer’s comment. We checked again all data and confirmed the no mistake.

Reviewer's comment

It will be better if the authors can add another Figure showing the overall model of their findings.

Our response

We demonstrated the overall model as a Graphic abstract.

Round 2

Reviewer 2 Report

The authors appropriately addressed my comments and revised the manuscript. I get the authors rebuttal but the lack of nutrient restriction/deprivation to 3T3L1 or MEF is still concerning. The effects of over/under expression of any molecular target in these cell lines simply indicates the effects of the target being manipulated and may have nothing to do with caloric restriction. For example, the revised statement that “We also demonstrated that CR upregulates Fgf21 via SREBP-1c in mice on both B6;129S6 and B6;129S6 backgrounds” (lines 515-516) is supported by Fig 1. I hope that you mean C57BL6 and B6 mice.

However, there are other claims that are not supported by the data. For example, the statement that “However, we found that CR-induced mitochondrial biogenesis is mediated by Srebp-1c only in WAT” (lines 563-564) are not supported by the data. The effects of SREBP1c OE on mitochondrial markers in 3T31 cells (Fig 2) may have nothing to do caloric restriction but could be independent effects of SREBP1c. This statement can be rephrased to something like “Srebp-1c may play a role in the CR-induced mitochondrial biogenesis in WAT”. Please tone down such statements extrapolating the effects from cell lines to in vivo CR effects in the rest of the discussion and abstract as I indicted in the previous review.

Author Response

Reviewer’s comment

The authors appropriately addressed my comments and revised the manuscript. I get the authors rebuttal but the lack of nutrient restriction/deprivation to 3T3L1 or MEF is still concerning. The effects of over/under expression of any molecular target in these cell lines simply indicates the effects of the target being manipulated and may have nothing to do with caloric restriction.

Our response

First of all, I want to say the followings. To support our previous findings obtained from Srebp-1c KO mice on a B6;129S6 background (), we performed the in vitro experiments predominantly in this study. We added the previous findings in detail in the 2nd paragraph in Introduction section.

Reviewer’s comment

For example, the revised statement that “We also demonstrated that CR upregulates Fgf21 via SREBP-1c in mice on both B6;129S6 and B6;129S6 backgrounds” (lines 515-516) is supported by Fig 1. I hope that you mean C57BL6 and B6 mice.

Our response

We apology our mistake. I corrected the statement.

Reviewer’s comment

However, there are other claims that are not supported by the data. For example, the statement that “However, we found that CR-induced mitochondrial biogenesis is mediated by Srebp-1c only in WAT” (lines 563-564) are not supported by the data. The effects of SREBP1c OE on mitochondrial markers in 3T31 cells (Fig 2) may have nothing to do caloric restriction but could be independent effects of SREBP1c. This statement can be rephrased to something like “Srebp-1c may play a role in the CR-induced mitochondrial biogenesis in WAT”.

Our response

The statement was misleading description. The finding was derived from our previous report. I added reference number. In addition, we added the comment in the 2nd paragraph in Introduction section as described above.

Reviewer’s comment

Please tone down such statements extrapolating the effects from cell lines to in vivo CR effects in the rest of the discussion and abstract as I indicted in the previous review.

Our response

We toned down our statement in the last paragraph in Discussion section.

This manuscript is a resubmission of an earlier submission. The following is a list of the peer review reports and author responses from that submission.

Round 1

Reviewer 1 Report

The study of Kobayashi and colleagues seeks to determine whether leptin signaling in WAT regulates expression of SREBP-1C, FGF21, and PGC-1a mRNAs and proteins and whether caloric restriction can also control those signals via SREBP-1C in WAT. Although some findings are interesting, the overall results don’t support their conclusions. Moreover, most results come from only 4 experiments or 4 mice, which appears to be not sufficient to find significant differences between the two groups.

Major comments:

  1. The number of experiments is too small, so it is very hard to determine whether there are significant differences between the groups. All figures should be presented as individual value plots.
  2. 2H and 3H: Why are there so much differences in mock experiments? It appears that deletion of Srebp-1c or Srebp-1a either downregulates or upregulates expression of proteins involved FA biosynthesis and mitochondrial biogenesis in controls. When compared the results obtained with Srebp-1C OE (Fig. 2H) with those from control experiments (Fig. 3H), there are huge increases in the protein expression. Are there any particular reasons for that? How do they get the results shown in Fig. 2A? In fact, the authors indicate that qRT-PCR of Srebp-1c is not possible due to too low expression of Srebp-1c mRNAs in 3T3-L1 cells.
  3. 5C: which bands are for pERK and ERK? Top or bottom? Where is the protein size band?
  4. 7: The authors need to show reduced protein expression of leptin receptors in leptin receptor shRNA-treated cells.
  5. 7C: These are relative expression. Why is Srebp-1a expression in control so low?

Minor comments:

  1. 2 and 3 labeling is not correct.      

Author Response

Major comments:

M1.The number of experiments is too small, so it is very hard to determine whether there are significant differences between the groups. All figures should be presented as individual value plots.

⇒In the revised manuscript, we remade all the graphs as the reviewer recommended. In the case of knockout (KO) mice experiments, we obtained the same results in mice on both B6;129S6 and C57Bl/6 backgrounds. We previously reported the results of Pgc-1α mRNA expression in Srebp-1c KO mice on a B6;129S6 background (Fujii et al. 2017). In addition, we showed the results of Fgf21 mRNA expression in Srebp-1c KO mice on a B6;129S6 background in Supplementary Figure S1 of the revised manuscript.

M2.① 2H and 3H: Why are there so much differences in mock experiments? It appears that deletion of Srebp-1c or Srebp-1a either downregulates or upregulates expression of proteins involved FA biosynthesis and mitochondrial biogenesis in controls. When compared the results obtained with Srebp-1C OE (Fig. 2H) with those from control experiments (Fig.3H), there are huge increases in the protein expression. Are there any particular reasons for that?

⇒The mock cell line presented in Figure 2 is not the same as the cell line presented in Figure 3. Moreover, exposure time is not always similar for each round of western blotting. Therefore, the expression pattern in Figure 2 appears slightly different from that shown in Figure 3.

M2.② How do they get the results shown in Fig. 2A? In fact, the authors indicate that qRT-PCR of Srebp-1c is not possible due to too low expression of Srebp-1c mRNAs in 3T3-L1 cells.

⇒Because intrinsic expression of Srebp-1c mRNA was very low in 3T3-L1 adipocytes, it was difficult to compare the low level of intrinsic Srebp-1c mRNAs derived from mock and Srebp-1a OE cell lines using real-time RT-PCR (Figure 3B). However, because the Srebp-1c OE cell line expresses a large amount of extrinsic Srebp-1c mRNA, we were able to compare the mRNA level of Srebp-1c in this cell line with the mock cell line using real-time RT-PCR analysis (Figure 2A), regardless of the very low intrinsic expression of Srebp-1c mRNA.

M3. 5C: which bands are for pERK and ERK? Top or bottom? Where is the protein size band?

⇒High and low molecular bands show ERK1 and ERK2, respectively. In the revised manuscript, we have indicated ERK1 and ERK2 with arrows.

M4. The authors need to show reduced protein expression of leptin receptors in leptin receptor shRNA-treated cells.

⇒We performed western blotting for leptin receptor using an antibody purchased from Santa Cruz Biotechnology (Ob-R, clone B-3; Catalog No. sc-8391, Dallas, TX, USA). However, because the protein level in white adipose tissue and 3T3-L1 adipocytes is very low compared with the liver, it is difficult to compare leptin receptor expression levels between mock and shLeptinR adipocytes.

M5. Figure 7C: These are relative expression. Why is Srebp-1a expression in control so low?

⇒We appreciated the reviewer pointing this out. In the revised manuscript, we remade this graph.

minor comments:

m1. 2 and 3 labeling is not correct.

⇒We appreciate the reviewer pointing this out. In the revised manuscript, we remade the graphs in Figures 2 and 3.

Reviewer 2 Report

Kobayashi et al. have made an extensive work, basically in vitro, in order to decipher the molecular mechanisms relating leptin signaling, SREBPs, FGF21, and PGC-1alpha with white adipose tissue expression associated with calorie restriction-induced remodeling. The general narrative is good and supported by previous research of the group, and the experimental procedures seem to be done with accuracy. The combination of different in vitro approaches of gene silencing and overexpression, and KO-derived cells, is also very interesting. Nevertheless, this reviewer has some concerns and comments, explained at the following.

Major points:

1- The title and the general conclusion do not seem to fit with the experiments done in this specific work, and seem to be more based on previous studies. Although the manuscript is written with a good narrative, the main focus given by the title and some parts of the discussion are confusing, since the results which are shown here are basically based on in vitro experiments with cell models and not with animals subjected to calorie restriction (such experiments were mainly done in previous works explained by the authors). The only figure (out of 7 with results) showing results of calorie restriction experiments is figure 1, basically showing that Pgc-1alpha and Fgf21 upregulation under calorie restriction is lost in Srebp-1c KO animals. The questions that the authors raise in order to further the knowledge on the molecular mechanisms relating leptin signaling, SREBPs, FGF21, and PGC-1alpha in white adipose tissue expression is sound, but the interpretation, especially in some parts of the discussion and in the title, is misleading. Although they can hypothesize about possible relations and the present results can help to understand previous results obtained in vivo, the authors must consider that some observations made in the in vitro experiment cannot be directly translated to the calorie restriction situation. Therefore, this reviewer suggests a re-thinking and re-writing of the paper, especially the discussion (and the title). E.g. the last paragraph of the discussion (conclusion) seems to be more based on previous works. Other fragments and sentences along the paper should be put into context. Consequently, some adjustments can also be made in the abstract.

2- Throughout the seven figures with results, it is stated that the "n" was 4. This is usually small. Although when working with KO animals this can be justified, it is not so clear for all the cell experiments (e.g. the use of MEFs allows the expansion of cells and opens the possibility to increase the number of cultures and wells or flasks used per culture). A better explanation (or justification) of the "n" of the experiments could be helpful. Moreover, for the culture experiments, it would be important to explain how many different cultures were done for each experiment and if, for instance, in each culture experiment there were triplicates (or more) for every condition or treatment. Regarding the MEFs, it is also important to know if different clones were used in each experiment (not all from the same embryo).

3- The Srebp-1c KO mice used to obtain MEFs are also the same Srebp-1c KO mice on a C57Bl/6 background? And the Fgf21 KO mice?

Minor points:

1- The use of English is very good, only check small mistakes, such as "the an" in line 28 (abstract).

2- The authors can explain the rationale of using MEFs instead of other models such as primary cultures of white adipose tissue.

3- Please, explain the rationale of using the C57Bl/6 background.

4- Line 138: explain here that PD1730741 is an FGFR inhibitor.

5-  Statistics are quite simple. Have the authors considered the possibility of two-way ANOVA in some cases or equivalent tests if parametric tests are not possible?

6- Check Y axis legends in figures 2 and 3, I-N (it should be "protein" instead of mRNA).

7- This reviewer has not found figure 8, which is cited at the end of the discussion.

Author Response

Major points:

M1. The title and the general conclusion do not seem to fit with the experiments done in this specific work, and seem to be more based on previous studies. Although the manuscript is written with a good narrative, the main focus given by the title and some parts of the discussion confusing, since the results which are shown here are basically based on in vitro experiments with cell models and not with animals subjected to calorie restriction (such experiments were mainly done in previous works explained by the authors). The only figure (out of 7 with results) showing results of calorie restriction experiments is figure 1, basically showing that Pgc-1alpha and Fgf21 upregulation under calorie restriction is lost in Srebp-1c KO animals. The questions that the authors raise in order to further the knowledge on the molecular mechanisms relating leptin signaling, SREBPs, FGF21, and PGC-1alpha in white adipose tissue expression is sound, but the interpretation, especially in some parts of the discussion and in the title, is misleading. Although they can hypothesize about possible relations and the present results can help to understand previous results obtained in vivo, the authors must consider that some observations made in the in vitro experiment cannot be directly translated to the calorie restriction situation. Therefore, this reviewer suggests a re-thinking and re-writing of the paper, especially the discussion (and the title). E.g. the last paragraph of the discussion (conclusion) seems to be more based on previous works. Other fragments and sentences along the paper should be put into context. Consequently, some adjustments can also be made in the abstract.

⇒We appreciate the reviewer pointing this out. Based on the reviewer’s suggestion, we changed the title and rewrote the abstract and last paragraph of the Discussion section. In addition, we changed the title from “Role of a leptin signaling/Srebp-1c/Fgf21/Pgc-1α axis in the caloric restriction-associated metabolic remodeling of white adipose tissue” in the original version to “Srebp-1c/Fgf21/Pgc-1α axis regulated by leptin signaling in adipocytes – possible mechanism of caloric restriction-associated metabolic remodeling of white adipose tissue” in the revised version. However, the main theme of our manuscript was to clarify the molecular mechanism in vitro of caloric restriction-associated metabolic remodeling of WAT observed in mice. Therefore, we do not want to change the main story of the major part of our manuscript.

M2. Throughout the seven figures with results, it is stated that the "n" was 4. This is usually small. Although when working with KO animals this can be justified, it is not so clear for all the cell experiments (e.g. the use of MEFs allows the expansion of cells and opens the possibility to increase the number of cultures and wells or flasks used per culture). A better explanation (or justification) of the "n" of the experiments could be helpful. Moreover, for the culture experiments, it would be important to explain how many different cultures were done for each experiment and if, for instance, in each culture experiment there were triplicates (or more) for every condition or treatment. Regarding the MEFs, it is also important to know if different clones were used in each experiment (not all from the same embryo).

⇒We appreciate the reviewer pointing this out. We obtained the almost same results in mice on both B6;129S6 and C57Bl/6 backgrounds. We previously reported the results of Pgc-1α mRNA expression in Srebp-1c KO mice on a B6;129S6 background (Fujii et al. 2017). In addition, we showed the results of Fgf21 mRNA expression in Srebp-1c KO mice on a B6;129S6 background in Supplementary Figure S1 of the revised manuscript. Therefore, the number of mice examined for each genotype and experiment in this manuscript is only four, but we considered the accuracy to be guaranteed.

⇒In our in vitro study using MEFs, we used four cell lines derived from four individual embryos. In the revised manuscript, we added a description of this in the figure legends of Figures 4 and 6. In our in vitro study of OE and KD cell lines, we performed real-time RT-PCR and western blotting using mRNAs and protein lysates, respectively, extracted from four separate dishes derived from the same cell line. As the degree of adipocyte differentiation is similar but not identical in each dish, we added a description of this limitation in the figure legend of Figure 2, 3, 5 & 7 in the revised manuscript.

⇒For most data, the alteration of protein levels was similar to that of mRNA levels. In addition, we performed experiments using both upregulated gene expression with OE cells and downregulated gene expression with KD or KO cells for each factor. Taken together, we confirmed our conclusion. We have added a description of this in line 412 to line 415 in the Discussion section of the revised manuscript.

M3. The Srebp-1c KO mice used to obtain MEFs are also the same Srebp-1c KO mice on a C57Bl/6 background? And the Fgf21 KO mice?

⇒MEFs were derived from embryos of Srebp-1c KO and Fgf21 KO mice on a C57Bl/6 background. We added a description of this in line 124 to line 125 in the Material and Methods section of the revised manuscript.

minor points:

m1. The use of English is very good, only check small mistakes, such as "the an" in line 28 (abstract).

⇒We appreciate the reviewer pointing this out. We have corrected the typing mistake in the revised manuscript.

m2. The authors can explain the rationale of using MEFs instead of other models such as primary cultures of white adipose tissue.

⇒Because SREBP-1a and -1c are derived from a single gene through the use of alternate transcription start sites that produce alternate forms of exon 1, which have very high homology (Shimomura et al., J Clin Invest. 99:838–845, 1997). For this reason, it is difficult to generate a Srebp-1c-specific KD cell line. Therefore, we used Srebp-1c KO MEFs. We have also extensive experience to harvest MEFs. Moreover, we can get the larger number of adipocytes derived from MEFs rather than that of primary culture adipocytes.

m3. Please, explain the rationale of using the C57Bl/6 background.

⇒We sometimes experience phenotypic difference among different mouse strains. Therefore, to confirm the data obtained in mice on a B6;129S6 background, we analyzed the effect of Srebp-1c in Srebp-1c KO mice on a C57Bl/6 background and compared both results. We added a description of this in line 204 to 211 in the Results section of the revised manuscript.

m2-4. Line 138: explain here that PD1730741 is an FGFR inhibitor.

⇒We appreciate the reviewer’s suggestion. We have added an explanation of PD1730741 in the revised manuscript.

m5. Statistics are quite simple. Have the authors considered the possibility of two-way ANOVA in some cases or equivalent tests if parametric tests are not possible?

⇒We appreciate the reviewer’s suggestion. We analyzed the data using two-way ANOVA in Figure 1 and 5, and have added a description of this in the revised manuscript.

m6. Check Y axis legends in figures 2 and 3, I-N (it should be "protein" instead of mRNA).

⇒We appreciate the reviewer pointing out our mistakes. We have corrected these legends in the revised manuscript.

m7. This reviewer has not found figure 8, which is cited at the end of the discussion.

⇒We appreciate the reviewer pointing out our mistake. We deleted the phrase “Figure 8” in the revised manuscript.